# The Effects of Biofeedback Training and Smartphone-Delivered Biofeedback Training on Resilience, Occupational Stress, and Depressive Symptoms among Abused Psychiatric Nurses

**DOI:** 10.3390/ijerph17082905

**Published:** 2020-04-22

**Authors:** Hsiu-Fen Hsieh, I-Chin Huang, Yi Liu, Wen-Ling Chen, Yi-Wen Lee, Hsin-Tien Hsu

**Affiliations:** 1School of Nursing, College of Nursing, Kaohsiung Medical University, No. 100, Shih-Chuan 1st Road, Kaohsiung 807, Taiwan; hsiufen96@gmail.com (H.-F.H.); jammi.huang@gmail.com (I.-C.H.); gn94yliu@kmu.edu.tw (Y.L.); 2Department of Medical Research, Kaohsiung Medical University Hospital, Kaohsiung Medical University, Kaohsiung 807, Taiwan; 3Department of Nursing, Tsyr-Huey Mental Hospital, Kaohsiung, 831 Taiwan; emma630902@yahoo.com.tw; 4Nursing Department, Kaohsiung Chang Gung Memorial Hospital, Kaohsiung 833, Taiwan; hikare2000@adm.cgmh.org.tw

**Keywords:** psychiatric nurses, workplace violence, biofeedback training, smartphone, depressive symptoms, resilience, occupational stress, heart rate variability

## Abstract

Psychiatric ward (PW) nurses are at a higher risk to encounter workplace violence than are other healthcare providers, and many interventions have been developed to improve their mental health. We compared the effectiveness of biofeedback training (BT) and smartphone-delivered BT (SDBT) interventions on occupational stress, depressive symptoms, resilience, heart rate variability, and respiration rate in a sample of abused PW nurses. This was a quasi-experimental study. Structured questionnaires were administered before and six weeks after the intervention. Data were collected from April 2017 to October 2017. A total of 159 abused PW nurses were randomly assigned to BT, SDBT, and control groups, and 135 of them completed all processes of our protocol, with the study consisting of 119 females (88.1%) and 16 males (11.9%) and their age range being from 22 to 59 with the mean age of 35.61 and a standard deviation of 8.16. Compared to the controls, both the BT and the SDBT intervention groups experienced significant improvements in depressive symptoms, resilience, and respiration rate; and the SDBT group experienced significant reductions in occupational stress. Considering the cost, accessibility, restrictions time and space, SDBT be used as an effective intervention in people with resilience or occupational stress.

## 1. Introduction

Workplace violence (WPV) is a global issue. The World Health Organisation [1] reported that WPV affects healthcare providers, which requires the attention of all medical institutions. The International Council of Nurses [2] also issued a position statement in which they condemn any form of violence against nurses. Healthcare professionals are at a higher risk of WPV than other workers [1,3]. Approximately 8% to 38% of healthcare providers experience WPV, and even more nurses are threatened or exposed to verbal aggression [1].

One study found that up to 99% of psychiatric ward (PW) healthcare providers reported that they had experienced verbal violence from patients or their families, and approximately 70% had been physically assaulted by patients in the past 12 months [4]. A similar study showed that PW nurses experienced an average incidence of verbal and physical violence of 0.6 and 0.19 times per nurse per week, respectively [5]. One study reported that physical and psychological violence were experienced annually by 55.7% and 82.1% of the PW nurses in Taiwan, respectively [6]. Moreover, PW nurses are more likely to experience violent attacks by patients or their families than nurses in other wards [7].

WPV is a cause of high stress, which might lead to physical and mental health problems among nurses who are already prone to high working stress. This increased stress might lead to nursing turnover [8] or negatively affect care quality [9]. Some studies found that WPV may lead to a variety of negative outcomes, including stress, job dissatisfaction, low self-esteem, fear, anxiety, poor quality of sleep, post-traumatic stress disorder, [10] or severe depression symptoms [11]. In addition, some studies found that nurses who are chronically exposed to a violent environment display an increased rate of taking antidepressants, antianxiety medicine, and sleeping pills [12,13].

Resilience is the process of adapting well in the face of adversity, trauma, tragedy, threats, stress, serious health problems, or workplace conflict—it means ‘bouncing back’ from difficult experiences [14]. Workplace resilience interventions have been shown to increase job satisfaction, work engagement, and resilience [15]; as well as decrease burnout and perceived stress [15,16,17,18]. Many interventions have been effective for enhancing one’s resilience, such as positive psychology, psychotherapy, biofeedback, meditation, and yoga [19]. Biofeedback has been used to help individuals recognise many physiological functions and manipulate their bodily systems with changes in heart rate variability (HRV) via regulating breathing [20]. Further, biofeedback also improves health, performance, and varied physiological changes (e.g., brain waves, muscle tone, skin conductance, heart rate, and respiration rate) [21]. In summary, the use of biofeedback interventions is effective in managing stress, anxiety, depression, and enhancing resilience [15,18]. In addition to computer-assisted biofeedback, there is a growing interest in the use of smartphone applications for biofeedback to promote better stress-management and relaxation [17,22]. In other words, smartphones can be used as a tool for biofeedback training (BT) because smartphones and the Internet are popular, conveniently accessible, and have an array of technological capabilities [17,23].

Several studies have examined the effects of BT in many populations; however, there is no associated study that has addressed abused PW nurses. The aim of this study was to compare the effectiveness of a BT and a smartphone-delivered BT (SDBT) intervention on the occupational stress, depressive symptoms, resilience, HRV, and respiration rate of abused PW nurses. We hypothesised that the BT intervention would be associated with the greatest improvements in occupational stress, depressive symptoms, resilience, HRV, and respiration rate relative to the SDBT intervention and the control condition owing to the benefits of face-to-face administration. We also hypothesised that the SDBT intervention would result in greater improvements in these variables as compared to the control group.

## 2. Methods

### 2.1. Design and Setting

We employed an experimental design with cluster random sampling—assigning eligible participants in the same hospital to either the BT intervention, the SDBT intervention, or the control group by a random number generator. Participants were queried regarding their ability to attend intervention sessions in particular time blocks. Owing to this randomisation method, self-selection into a particular intervention could not occur.

### 2.2. Participants

A total of 159 PW nurses were recruited from three hospitals. Before collecting data, meetings with the managers and head nurses of each participating hospital were arranged to present this study. Then, the principal investigator explained the purpose and process to the PW nurses in each ward and invited them to participate in this study. The inclusion criteria were nurses who (1) had worked for at least three months, (2) had experienced any type of workplace violence exerted by patients or their families in the past 12 months, and (3) owned a smartphone. The exclusion criterion for this study was participants who had existing mental disorders before experiencing WPV, such as depression, generalised anxiety disorder, etc.

### 2.3. Research Procedure

All participants were asked to take a resilience-enhancing course for two hours before being divided into the BT, SDBT, or control group. The BT group received one 60-min session of BT weekly for six weeks. The SDBT group received MP4-video guided biofeedback self-training weekly for six weeks. The control group did not receive the BT or SDBT intervention. The abused PW nurses’ resilience, depressive symptoms, occupational stress were measured at two points. Physiological measurements and self-report data were collected pre-test (week 0) and post-test (week 6), and all participants’ physiological indices were measured by the same machine (ProComp InfinitiTM) to avoid measurement bias. The participants in the control group were invited to receive the BT or SDBT intervention after the post-intervention measurement if they wished. Data were collected from April 2017 to October 2017.

#### 2.3.1. Biofeedback Training

The BT protocol was based on Lin and colleagues [24], which included self-guided muscle relaxation, diaphragmatic breathing, paced breathing, pursed-lips breathing, and real-time respiratory sinus arrhythmia (RSA) biofeedback over a 60-min session, weekly for six weeks. BT incorporates aspects of real-time RSA-biofeedback and shorter meditation practices. The BT has been studied previously in traumatic patients with positive results [23].

#### 2.3.2. Smartphone-Delivered Biofeedback Training

Based on the conception of BT, every participant in the SDBT group was provided with an MP4 video file containing guided shorter meditation practices and the processes of real-time biofeedback once a week for six weeks to enhance their resilience. This was an app-directed, self-managed protocol. All participants were tracked and checked that they completed the training by cloud service.

### 2.4. Data Collection and Ethical Considerations

This study was approved by the Institutional Review Board (no. KMUHIRB-E(II)–20170101) of the appropriate Kaohsiung Medical University hospital before the study was conducted. The principal investigator approached eligible participants individually and invited them to participate, explained the purpose of the study, and asked them if they were willing to participate. Participants were also informed that their participation was voluntary, and they had the right to terminate their participation at any time without reason. All participants were assured of data confidentiality and anonymity. Written informed consent was obtained from every participant. Participants were offered NT $400 (approximately US $12–14) as gratitude for study completion.

### 2.5. Instruments

Various measures were used pre-intervention including a demographic basic information sheet, the rehabilitation strength chart, the simplified health scale, the Centre for Epidemiologic Studies Depression scale (CES-D), the Occupational Stress Indicator-2 (OSI-2), the Resilience Scale (RS), HRV, and respiration rate.

#### 2.5.1. Physiological Measurements

We used ProComp InfinitiTM software (Thought Technology Ltd., Montreal, Canada), which was installed on a laptop, and an electrocardiogram (for measuring HRV, such as standard deviation of normal to normal (SDNN), low frequency (LF), and high frequency (HF)) and respiration sensors, to obtain real-time feedback regarding respiration rate, breathing patterns (paced breathing at 5 or 6 breaths per minute), RSA amplitude, and HRV indices. The HRV indices as the outcome variables were obtained at pre- and post-intervention. Concerning the meaning of HRV indices, SDNN and total power of HRV indicated the overall HRV, LF was most likely affected both by cardiac SNS and PNS, and HF was an index of PNS activation [25,26].

##### Centre for Epidemiological Studies Depression

We measured depressive symptoms using the Chinese version of the Centre for Epidemiological Studies Depression (CES-D), with 20 items taken from Chien and Cheng [27]. Each item was measured with a 4-point Likert scale to access levels of agreement to each statement. Total scores ranged from 0 to 60, and the cut-off point was ≥ 15. Cronbach’s alpha was 0.86 in Chien and Cheng [27]. The CES-D is widely used to assess depression in many fields of research, including the relationship between WPV and depression [28]. The Cronbach’s alpha of the CES-D was 0.87 in the present study.

##### Resilience Scale

The Resilience Scale (RS) was developed by Friborg and colleagues [29] and the Chinese version was established by Wang and Chen [28] with satisfactory validity and reliability. The 29-item scale measures intrapersonal and interpersonal protective resources that may facilitate an individual’s adaptability to and tolerance for stress and adverse life events. The scale comprises five components: personal strength, social competence, structured style, family cohesion, and social resources. The total scores of the RS range from 29 to 203, with higher scores indicating a higher level of resilience. A confirmatory factor analysis allowing all factors to correlate indicated a satisfactory fit. The 3 to 4-week test–retest reliability of intra-class correlation coefficient was 0.89, and Cronbach’s alphas were 0.92, 0.85, 0.85, 0.83, and 0.87 for the five factors, respectively [30]. The Cronbach’s alpha of the entire RS was 0.94 in the present study.

##### Occupational Stress

The revised Chinese version of The Occupational Stress Indicator (OSI-2) [31] was implemented in our study, and it consists of 40 items for measuring eight subscales of occupational stress with 6-point scales. The scale contains eight subscales: workload, relationship, home/work balance, managerial role, personal responsibility, hassles, recognition, and organisation climate. All subscales are summed and higher scores indicate greater occupational stress. The Cronbach’s alphas for this scale were 0.83 [31] and 0.97 in the present study.

### 2.6. Data Analysis

Descriptive analysis, t-tests, and the nonparametric Kruskal-Wallis test were performed using SPSS 19.0 (SPSS Inc, Chicago, IL, USA). Descriptive analysis was generated from the demographic characteristics to describe participants’ age, education, marital status, and religious beliefs. T-tests were conducted to examine depressive symptoms, resilience, occupational stress, SDNN, LF, HF, and respiration rate between pre- and post-test in the three groups. In addition, the nonparametric Kruskal-Wallis test with post-hoc comparisons were used to examine the group differences on depressive symptoms, resilience, occupational stress, HRV (SDNN, LF, and HF) and respiration rate among the three groups. All tests were two-tailed and *p* < 0.05 was considered significant.

## 3. Results

### 3.1. Demographics and Outcomes at Baseline

The rates of attrition for BT, SDBT and the control group were 11.3%, 26.4%, and 7.5%, respectively. The average age in BT group was significantly higher than in the SDBT group (*p* = 0.002), and there were no significant differences in baseline measures or demographics among these three groups.

Six weeks later, eight of the abused PW nurses had left their positions, and 135 abused PW nurses completed the post-test. Participants’ characteristics are shown in Table 1.

### 3.2. BT, SDBT, and Control Group Changes Over Time from Baseline to the end of Week 6

The changes in the three groups from baseline to the end of week 6 are shown in Table 2. First, concerning BT, participants had significantly decreased depressive symptoms and increased resilience. In addition, respiration rates decreased significantly pre- to post-intervention for the experimental conditions; however, there were no significant differences in occupational stress, SDNN, LF, or HF. Second, concerning SDBT, participants had significantly decreased depressive symptoms and occupational stress, and increased resilience. In addition, respiration rates decreased significantly pre- to post-intervention; however, there were no significant differences in SDNN, LF, and HF.

### 3.3. The Kruskal-Wallace Testsresults for the Mean Differences in Depressive Symptoms, Occupational Stress, and Resilience among BT, SDBT, and the Control Group

The distribution of the depressive symptoms, occupational stress, and resilience were skewed, and we used the Kruskal-Wallis test to compare mean ranks. The Kruskal Wallace nonparametric (pre-intervention and post-intervention) and post-hoc tests comparison revealed that there were no significant difference among the BT group, SDBT group and the control group in depressive symptoms, and that the BT and SDBT groups had significantly higher resilience than did the control group. In addition, the SDBT group had significantly lower occupational stress than did the control group. However, there were no significant differences among the three groups in SDNN, LF, HF values and respiration rate (Table 3).

## 4. Discussion

This randomised controlled trial compared the effects of a BT intervention, SDBT intervention, and control group on depressive symptoms, resilience, occupational stress, HRV (SDNN, LF, and HF), and respiration rate in abused PW nurses. Both the BT and the SDBT intervention groups experienced significant improvement in depressive symptoms, resilience, and respiration rate following the intervention. The SDBT intervention group experienced significant reductions in occupational stress. The control group experienced improvement only in depressive symptoms, and they experienced no significant change in other variables. This outcome might result from the effect of the 2-h resilience-enhancing course at the beginning of this study, and a similar result was obtained by Haracz and Roberts [15]. Furthermore, we found that both BT and SDBT were effective interventions for improving depressive symptoms, resilience, and respiration rate; and the SDBT intervention may also confer broader benefits such as reducing occupational stress.

Depressive symptoms are not uncommon in psychiatric workers [32], and these depressive symptoms are more common among those PW nurses who encounter violence versus those who do not [33]. Abused PW nurses require effective interventions for improving depressive symptoms, and we found that both the BT and SDBT groups had a greater reduction of depressive symptoms than did the control group, with significant difference. This result was congruent with previous studies [34,35]. Moreover, The Kruskal Wallace nonparametric (pre-intervention and post-intervention) and post-hoc tests comparison revealed that there was no significant difference among the BT group, SDBT group and the control group in depressive symptoms, whereas the BT and SDBT groups had greater reduction in depressive symptoms than the control group., which was not consistent with a previous study [17]. Our results suggested that BT and SDBT had a contribution on improving depressive symptoms, compared to the control group, without significant difference. An insufficient duration of the intervention might be the reason for no significant difference among these three groups.

Resilience is an important factor associated with psychological health in abused PW nurses [36], and resilience can further reduce depressive symptoms [37]. Our results showed that both the BT and SDBT interventions can significantly strengthen abused PW nurses’ resilience, as compared to controls, and such results were also found in similar previous studies [22,33].

We found that the SDBT intervention, but not the BT intervention group or the control group, produced significant reductions in occupational stress, and this result was contrary to our hypothesis, namely that we hypothesised the BT would be more effective than SDBT. In addition, the Kruskal Wallace H test and post-hoc tests comparison also revealed that the SDBT group had significantly lower occupational stress than did the control group. One study showed that resilience-based and relaxation music-based smartphone applications was better than a control group in improving perceived stress in workers [17], and a similar result was also obtained in our study. BT can be provided only in the hospital which was a relatively a stressful place to nurses, and SDBT was usually carried out during nonworking hours when individuals felt relatively less stress. This might explain our result that the SDBT was more effective than BT in reducing occupational stress. Psychiatric nursing is a high-stress job [38,39], and abused PW nurses have especially high stress in the workplace. Abused PW nurses need appropriate interventions to enhance their resilience for reducing stress. Unmanaged occupational stress is detrimental to both psychological and physical functioning [40], and it can lead to job exhaustion. Resilience-strengthening interventions have been used in several occupational settings to relieve workers’ stress response [22]. Further investigation into this topic is worthwhile because smartphone applications may be more feasible to implement than other interventions, which are restricted by time and place [23].

Both BT and SDBT interventions slowed participants’ respiration rates significantly. In addition, the SDBT group had a significantly lower respiration rate than did the control group. In our study, BT and SDBT guided participants to regulate their breathing, and the subsequent breathing patterns can lead to increased parasympathetic activity [41], which is associated with calming and relaxing effects [42]. The findings of this study support the hypothesis that BT is an effective intervention that restores parasympathetic activation among abused PW nurses. A similar result was also obtained among manufacturing operators [43,44].

In sum, our results indicated that the BT and SDBT interventions can improve depressive symptoms, resilience, and respiration rate, and that the SDBT intervention provided an additional benefit—reducing occupational stress. Smartphones provide a range of possibilities for helping individuals recover from stressful experiences, such as leisure or social networks. In addition, a smartphone is usually used during nonworking hours when individuals feel relatively less stress as compared to working hours. In our study, the time that participants decided to receive the SDBT intervention was when they were not working, which might be the reason that the SDBT intervention was as effective as BT in improving resilience, depressive symptoms, and respiration rate. In addition, SDBT is more effective in reducing occupational stress than BT, which was performed in the hospital (i.e., a stressful environment). This is a key study limitation. Another limitation is that we only utilised a 6-week intervention design; thus, further longitudinal studies are needed.

In the future, the application of SDBT can be extended to all employees with stress, and if BT can be available anytime in the community as with SDBT, then it is worth comparing the effectiveness of communal BT and SDBT on reducing occupational stress. In addition to BT and SDBT interventions, we suggest that hospital managers should provide PW nurses with continuing education, social support and relevant resources to reduce their occupational stress. Furthermore, we suggest that policy makers must re-examine nursing management policies and nursing practice-related factors, for establishing a safer environment, offering resources, enhancing support systems and improving nursing shortage for PW nurses to live a healthy lifestyle without excessive workload and occupational stress.

## 5. Conclusions

Both BT and SDBT can improve abused PW nurses’ depressive symptoms, resilience, and paced breathing. In addition to these strengths, SDBT can also reduce occupational stress. Furthermore, compared with BT, SDBT has the following advantages: low cost, portable and readily accessible, as well as having limited time and space restrictions. We suggest that SDBT be used as an effective intervention in people with depressive symptoms or occupational stress, regardless of their experienced with WPV.

## Figures and Tables

**Table 1 ijerph-17-02905-t001:** Demographics and outcomes at baseline.

Variable	Total (*n* = 135) *M(SD)/n(%)*	BT (*n* = 49) *M(SD)/n(%)*	SDBT (*n* = 47) *M(SD)/n(%)*	Control (*n* = 39) *M(SD)/n(%)*
Age (years)	35.61 (8.18)	38.45 (9.23)	32.21 (6.36)	35.61 (7.47)
Sex (% female)	119 (88.10)	43 (87.76)	29 (61.70)	38 (97.43)
Married/Partnered (%)	80 (59.30)	35 (71.42)	19 (40.43)	26 (66.67)
Education (% over college)	105 (77.78)	34 (69.39)	39 (82.98)	32 (82.05)
Religious Beliefs (% yes)	87 (64.40)	40 (81.63)	18 (38.30)	29 (74.35)
Depressive Symptoms	14.96 (7.56)	14.47 (8.26)	15.08 (7.13)	15.35 (7.31)
Occupational Stress	61.96 (36.15)	63.45 (34.82)	58.36 (30.87)	63.39 (41.44)
Resilience	150.11 (25.98)	153.98 (26.58)	143.13 (26.28)	151.96 (24.56)

Note. BT = biofeedback training group; SDBT = smartphone-delivered biofeedback training group; Control = control group.

**Table 2 ijerph-17-02905-t002:** Changes in the biofeedback training (BT), smartphone-delivered biofeedback training (SDBT), and control group from baseline to the end of week 6.

Groups	BT (*n* = 49)			SDBT (*n* = 47)			Control (*n* = 39)		
Variable	Baseline *M (SD)*	6-Week *M (SD)*	*p*	Baseline *M (SD)*	6-Week *M (SD)*	*p*	Baseline *M (SD)*	6-Week *M (SD)*	*p*
DS	14.47 (8.26)	8.72 (6.10)	< 0.001 ***	15.08 (7.13)	8.13 (6.33)	< 0.001 ***	15.34 (7.31)	12.29 (6.95)	< 0.001 ***
RS	153.98 (26.58)	164.15 (23.16)	< 0.001 ***	143.13 (26.28)	158.77 (19.20)	< 0.001 ***	151.9 (24.56)	153.67 (23.75)	0.321
OS	63.45 (34.82)	55.55 (30.73)	0.072	58.36 (30.87)	42.97 (37.91)	0.013 *	63.38 (41.44)	62.18 (40.27)	0.723
SDNN (ms)	48.02 (26.21)	44.48 (23.87)	0.296	55.74 (33.93)	54.94 (32.68)	0.750	37.98 (18.33)	52.10 (81.56)	0.207
LF (ms^2^)	249.50 (215.20)	364.00 (416.66)	0.494	759.96 (346.72)	737.77 (343.44)	0.518	312.88 (144.30)	455.45 (442.49)	0.211
HF (ms^2^)	349.73 (362.29)	410.58 (416.71)	0.609	681.86 (593.83)	625.15 (597.35)	0.063	356.48 (226.16)	858.28 (716.50)	0.167
RR	14.72 (2.63)	13.84 (2.84)	<0.001 ***	16.02 (2.33)	14.73 (2.87)	0.002 *	15.19 (2.36)	15.34 (2.57)	0.616

Note. BT = biofeedback training; SDBT = smartphone-delivered biofeedback training; DS = depressive symptoms; OS = occupational stress; RS = resilience; SDNN = standard deviation of normal to normal; LF = low frequency; HF = high frequency; RR = respiration rate; *** *p* < 0.001, * *p* < 0.05.

**Table 3 ijerph-17-02905-t003:** The Kruskal Wallace nonparametric and post-hoc tests results for the mean differences *n* depressive symptoms, occupational stress, and resilience among the three groups.

Variable	BT (*n* = 49) Mean Rank	SDBT (*n* = 47) Mean Rank	Control (*n* = 39) Mean Rank	*X^2^* (K-W)	*p*	Post-hoc	Test Statistic	*p*
DS	65.82	60.13	76.36	4.002	0.135	---	4.002	0.135
RS	74.31	82.14	50.69	15.955	< 0.001	3-1 3-2 1-2	−23.62 −31.45 −7.83	0.003 * < 0.001 *** 0.355
OS	64.94	56.49	80.10	8.361	0.015	2-1 2-3 1-3	8.45 23.62 15.17	0.319 0.005 * 0.058
SDNN (ms)	66.02	69.41	67.37	0.162	0.922	---	0.162	0.922
LF (ms^2^)	61.65	64.08	75.71	3.543	0.170	---	3.543	0.170
HF (ms^2^)	68.37	63.13	70.16	0.749	0.688	---	0.749	0.688
RR	69.38	56.26	73.34	4.961	0.084	---	4.961	0.084

Note. DS = depressive symptoms; RS = resilience; OS = occupational stress; RR = respiration rate; BT = Biofeedback group; SDBT = Smartphone-delivered biofeedback group; K-W test = Kruskal-Wallace test; 1 = BT group; 2 = SDBT group; 3 = Control group; *** *p* < 0.001, * *p* < 0.05.

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
