# Peer review of "The Effects of Biofeedback Training and Smartphone-Delivered Biofeedback Training on Resilience, Occupational Stress, and Depressive Symptoms among Abused Psychiatric Nurses"

_ijerph, 2020, doi:10.3390/ijerph17082905_

Round 1
Reviewer 1 Report
The abbreviation SDBT, SDB is mixed in the text. You can unify the two words together.
I recommend you to add to the discussion why SDBT is more effective than BT unlike the hypothesis.
Author Response
1)The abbreviation SDBT, SDB is mixed in the text. You can unify the two words together.
1)A: Thank you for your recommendation, we have unified the SDBT and SDB training into “SDBT” (Line 230).
2) I recommend you to add to the discussion why SDBT is more effective than BT unlike the hypothesis.
2)A: Thank you for your recommendation, we have added our discussion on the reason why SDBT is more effective than BT unlike the hypothesis (Lines 300-303).

Reviewer 2 Report
Occupational injuries by nurses are indeed a worth issue to explore. It makes this article practical. However, based on academic requirements, I made the following review comments.
- The representation in Figure 1 needs clarification or correction by the authors. My suggestions are as follows:
- Figure 1 is an extension of Table 2. All the contents to be explained in Fig. 1 are shown in Table 2. Based on the article's streamlining principles, the authors are asked to think carefully about the necessity of Figure 1.
- If the authors think that Figure 1 is necessary, the authors should think about the way in which Figure 1 is presented to assist readers in reading. The following is the proposed amendment to Figure 1:
- The four figures in Figure 1 should be numbered, and the content of each small figure is explained below the figure. With the current picture presentation, readers cannot understand.
- Is the presentation of Figure 1 correct using a line chart? If you use a line chart, the content of the X axis is in an orderly relationship (so you can connect the points with lines). However, the four images in Figure 1 do not seem to have the above-mentioned relationship, so it is recommended to amend them. I recommend using bar charts.
- Please write the unit numbers on the vertical axis on the same side. In addition, authors should also mark the units on the vertical axis.
- Authors should add the title of the horizontal axis.
- For the reader's understanding, please note the significance (shown with an asterisk) on Figure 1.
- Based on graphics streamlining principles, please delete unnecessary horizontal lines. In addition, lines should appear on the vertical axis.
- The expression in Figure 3 is unclear, which will make the reader unable to understand. For Table 3, I make the following suggestions:
- Please indicate the meaning of the numbers in the table (including the numbers in parentheses).
- Please indicate the meaning of the numbers "1, 2, 3" in the "Post-hoc" field.
- "p" is lowercase and italic.
- The "t" in "t-tests" is lowercase but not italic.
- In the study of depressive symptoms, the three groups of control, BT, and SDBT were all significant (Table 2), but in Table 3, the comparison of BT and control was not significant ("Post-hoc" column). This result shows that BT has no effect on depressive symptoms. Although the authors explained in the third paragraph (line 285-295) of the discussion, the authors did not explain why BT is not significant in depressive symptoms. In addition, there is no in-depth discussion on comparative research in the past. I think that the wording of "this result was similar to a previous study" is insufficient for discussion. For discussion of the results, authors should state the same reasons as in previous studies. More importantly, the authors should explain why they are different from previous research. This discussion is clearly a deficiency of this article and should be supplemented; otherwise it will seriously reduce the contribution of this article. The same situation occurs in the results and discussion of occupational stress. The authors are requested to comment.

Author Response
1-1) The representation in Figure 1 needs clarification or correction by the authors. My suggestions are as follows:
Figure 1 is an extension of Table 2. All the contents to be explained in Fig. 1 are shown in Table 2. Based on the article's streamlining principles, the authors are asked to think carefully about the necessity of Figure 1. If the authors think that Figure 1 is necessary, the authors should think about the way in which Figure 1 is presented to assist readers in reading. The following is the proposed amendment to Figure 1:
The four figures in Figure 1 should be numbered, and the content of each small figure is explained below the figure. With the current picture presentation, readers cannot understand. Is the presentation of Figure 1 correct using a line chart? If you use a line chart, the content of the X axis is in an orderly relationship (so you can connect the points with lines). However, the four images in Figure 1 do not seem to have the above-mentioned relationship, so it is recommended to amend them. I recommend using bar charts.
1-1)A: Thank you for your recommendation, we have removed the Figure 1.
1-2) Please write the unit numbers on the vertical axis on the same side. In addition, authors should also mark the units on the vertical axis. Authors should add the title of the horizontal axis. For the reader's understanding, please note the significance (shown with an asterisk) on Figure 1. Based on graphics streamlining principles, please delete unnecessary horizontal lines. In addition, lines should appear on the vertical axis.
1-2)A: Thank you for your recommendation, we have removed the Figure 1.
2. The expression in Figure 3 is unclear, which will make the reader unable to understand. For Table 3, I make the following suggestions:
2-1) Please indicate the meaning of the numbers in the table (including the numbers in parentheses).
2-1)A: Thank you for your recommendation, the meaning of the numbers in the Table 3 were added to the note below this table.
2-2) Please indicate the meaning of the numbers "1, 2, 3" in the "Post-hoc" field.
2-2)A: Thank you for your recommendation, we have added the meaning of each number in Table 3.
3-1) "p" is lowercase and italic.
3-1)A: Thank you for your recommendation, we have revised the letter "p" to be lowercase and italic.
3-2) The "t" in "t-tests" is lowercase but not italic.
3-2)A: Thank you for your recommendation, We have revised the " t " to be not italic.
4-1) In the study of depressive symptoms, the three groups of control, BT, and SDBT were all significant (Table 2), but in Table 3, the comparison of BT and control was not significant ("Post-hoc" column). This result shows that BT has no effect on depressive symptoms. Although the authors explained in the third paragraph (line 285-295) of the discussion, the authors did not explain why BT is not significant in depressive symptoms.
4-1)A: Thank you for your recommendation. The nonparametric Kruskal Wallace non parametric (pre-intervention and post-intervention) and post hoc tests comparison revealed that there were no significant difference among the BT group, SDBT group and the control group in depressive symptoms, whereas the BT and SDBT groups had greater reduction in depressive symptoms than the control group (Lines 249-259 and Table 3).
4-2) In addition, there is no in-depth discussion on comparative research in the past. I think that the wording of "this result was similar to a previous study" is insufficient for discussion. For discussion of the results, authors should state the same reasons as in previous studies. More importantly, the authors should explain why they are different from previous research. This discussion is clearly a deficiency of this article and should be supplemented; otherwise it will seriously reduce the contribution of this article. The same situation occurs in the results and discussion of occupational stress. The authors are requested to comment.
4-2)A: Our results suggested that BT and SDBT had contribution on improving depressive symptoms, compared to the control group without significant difference. Insufficient duration of intervention might be the reason of no significant difference among this three groups (Lines 290-293).

Reviewer 3 Report
See attached file.

Author Response
Review of “The effect of biofeedback training and smartphone-delivered biofeedback training on resilience, occupational stress, and depressive symptoms among abused psychiatric nurses” This manuscript adds important information for strategies to reduce stress, depressive symptoms, and build resilience among psychiatric nurses. It appears to fill a gap in the literature with this population related to these topics. Please see my major comments below:
1)The Introduction provided a succinct review of the issue and the gap in the literature for this population. For the Methods section, since this paper was focused on occupational stress, I am curious why the nurses were included if they experienced ANY type of violence? Does this include violence at home or violence in relationships? If so, this sets up a different dynamic I would think as the type of violence could be home/relationship only, home/relationship and work, or perhaps work only and may need much greater intervention. Also, throughout the paper the term “abused nurses” is used but not sure where the abuse happened and how or if that makes a difference. This needs to be explained in greater detail.
1) A: Thank you for your recommendation, we have corrected our had “Methods section” that all of our participants experienced any type of workplace violence exerted by patients or their families. We only included those participants who suffered from workplace violence which was exerted by patients or their families (Lines 92-93).
2-1) For the physiological measurements, the acronyms need to be better explained, as does SDNN. The readers may not be familiar with these terms pertaining to cardiac care. For the Results section, while it is true that T-tests were significantly different for several of the measures from baseline to post readings at six weeks, little interpretation of the ANOVA is done.
2-1)A: Thank you for your recommendation, we have explained each of the acronyms in the manuscript (Lines 135-136, and 139-141).
2-2)The ANOVA shows, for example, that occupational stress was not significantly different among the groups. However, in the Results and Discussion this is not mentioned yet the significance of SDBT and occupational stress at six weeks is discussed in detail. While the results of the T-tests do show a significant change, as stated, this was not upheld with the ANOVA test.
2-2)A: Thank you for your recommendation. The distribution of the occupational stress was skewed, and we used the nonparametric Kruskal-Wallis test to compare mean rank, and it revealed that the SDBT group had significantly lower occupational stress than did the control group. We have added some relevant descriptions in the “discussion” (Lines 249-257, 258-293, and 300-303).
2-3)Also, curiously, the significant change in occupational stress in the SDBT group was not mentioned in section 3.2 on page 5 when discussing the results for the T-tests?
2-3A: Thank you for your recommendation, we have added the significant change in occupational stress in the SDBT group (Line 246).
3-1)In the Results and Discussion sections, I believe more information should be provided as to next steps for this research (more sophisticated designs including longitudinal studies) and how to make it more generalizable.
3-1)A: Thank you for your recommendation, we have added some information in the section of discussion regarding further study (Lines 335-343 ).
3-2)Also, there is no discussion as to how psychiatric facilities, themselves, may change to support a better work environment for staff. Could staffing strategies be changed, should there be greater stress management opportunities offered, etc. While the biofeedback and Smartphone application may be effective, changes should be made in a more holistic manner so that interpersonal, organization, community, and policy changes and considerations are included. I would recommend references to these components be included.
3-2)A: Thank you for your recommendation, we have added some discussion about how to reduce PW nurses’ workload and their occupational stress (Lines 335-343 ).
4)This manuscript was written well however I could not find reference #20 in the text. References in the text and list need to be carefully matched.
4)A: Thank you for your recommendation, we have corrected the reference number (Line 63).

Round 2
Reviewer 2 Report
The authors have revised and supplemented the information in the article. There are no other problems.